# Effectiveness of a Complementary Telehealth Education Program as a Preventive Treatment for Chronic Migraine: A Randomized Pilot Study

**DOI:** 10.3390/jcm13226825

**Published:** 2024-11-13

**Authors:** Paula Cordova-Alegre, Pablo Herrero, Sonia Santos-Lasaosa, Maria Pilar Navarro-Perez, Beatriz Carpallo-Porcar, Sandra Calvo, Carolina Jimenez-Sanchez

**Affiliations:** 1Department of Physical Therapy, Universidad San Jorge, 50830 Zaragoza, Spain; pcordova@usj.es (P.C.-A.); bcarpallo@usj.es (B.C.-P.); cjimenez@usj.es (C.J.-S.); 2Instituto de Investigación Sanitaria (IIS) Aragón, 50009 Zaragoza, Spain; ssantos@salud.aragon.es (S.S.-L.); mapinape@hotmail.com (M.P.N.-P.); sandracalvo@unizar.es (S.C.); 3Department of Physiatry and Nursing, University of Zaragoza, 50001 Zaragoza, Spain; 4Hospital Clínico Universitario Lozano Blesa, 50009 Zaragoza, Spain

**Keywords:** chronic migraine, telehealth, therapeutic education, self-management

## Abstract

**Background/Objectives:** Chronic migraine (CM) is a neurological disorder that causes significant disability, loss of productivity, and economic burden. Preventive treatments, including pharmacological and educational interventions, are crucial for managing CM effectively. The aim of this study was to analyze whether adding a therapeutic telehealth education program (TTEP) to pharmacological treatment achieved a greater reduction in the number of headache days experienced by patients with CM. **Methods:** A randomized, double-blind, controlled pilot study with two parallel groups was performed. Patients with a diagnosis of CM and who were being treated with Botulinum Toxin were randomly assigned to either the EG (therapeutic education program about the neuroscience of pain, migraine, pain strategies, sleep habits, exercise, nutrition, postural habits, and relaxation strategies) or CG (general health recommendations with no specific content about migraine). The intervention lasted a total of eight weeks and was delivered via a telehealth application (APP). Headache frequency, migraine frequency, pain intensity, headache impact, allodynia, fear of movement, pain catastrophizing, chronic pain self-efficacy, anxiety and depression, sleep quality, and sedentary lifestyle were measured at baseline (M0), one month after the intervention started (M1), at the end of the intervention (M2), and one month after the intervention was completed for follow-up (M3). **Results:** In total, 48 patients participated. There were differences between the groups in the following outcomes in favor of EG for headache frequency at the one-month follow-up (*p* = 0.03; d = 0.681); chronic pain self-efficacy at post-treatment (*p* = 0.007; d = 0.885) and at the one-month follow-up (*p* < 0.001; d = 0.998); and sleep quality at post-treatment (*p* = 0.013; d = 0.786) and at the one-month follow-up (*p* < 0.001; d = 1.086). No differences existed between the groups for the other outcomes examined (*p* < 0.05). **Conclusions:** The use of TTEP reduced the number of headache days, improved sleep quality, and increased self-efficacy in managing pain. This pilot study suggests that the addition of a specialized TTPE to pharmacological treatments may be more effective than a general health recommendation program for migraine.

## 1. Introduction

Chronic migraine (CM) is a neurologic disorder associated with considerable disability, loss of productivity, and a profound economic burden, worldwide. It greatly impacts the patient, interfering with their physical function and psychological well-being [1,2,3].

The prevalence of CM varies according to the different studies and the diagnostic criteria used. Worldwide prevalence has been estimated to be between 1.4% and 2.2% of the population. It is more prevalent in Europe, followed by the Western Pacific, and then the Americas. CM is much more common in women and is one of the most common reasons for consultation with the general neurology service, since it affects 5–10% of patients in Europe [2,4,5].

According to the International Classification of Headache (ICDH-III), CM is diagnosed when a patient has experienced headaches on 15 or more days per month over the previous three months, with at least eight of those days fulfilling the criteria for migraine without aura or showing improvement with a triptan or ergot treatment [4,6,7].

Patients with CM also have central sensitization, which occurs due to the continuous stimulation of the nociceptors in the neurons of the trigeminal increased spinal nucleus and the thalamus. It manifests with cutaneous allodynia (CA). CA is usually limited to the cephalic area but may extend to extra-cephalic areas, being present in 80% of patients with CM. In addition, CM is associated with increased disability, medical and psychiatric co-morbidities, increased use of health care resources, and sleep disorders [8,9,10,11,12,13,14,15].

Therefore, preventive treatment is important in the management of patients with CM, which is not intended to prevent the disease, but to reduce the frequency of migraine episodes. The preventive treatment includes pharmacological and non-pharmacological interventions for pain self-management [16,17,18]. Non-pharmacological interventions aim to help patients manage their pain and symptoms more effectively. It has been shown that patients’ beliefs and previous knowledge have a great influence on the preventive treatment of chronic pain. For this reason, health education has proven to be an effective strategy for treating CM [19].

Health education refers to education provided by healthcare professionals with the aim of equipping patients with the necessary knowledge to manage their disease, improve their quality of life, and prevent complications. It also empowers patients with tools for self-management, adaptation to their chronic condition, and coping strategies to achieve maximum autonomy. Studies of chronic migraine (CM) patients have demonstrated that education focused on pain neuroscience, the physiological and psychological aspects of migraine, and a biopsychosocial approach—including relaxation techniques, lifestyle modifications, sleep hygiene, diet education, exercise therapy, self-regulation, and relaxation—is effective in the preventive management of migraine [19,20,21].

Telerehabilitation is the use of communication and technology to provide remote care to patients. It can be performed in different modalities: synchronous, that is, simultaneously, or asynchronous, that is, the clinician can prescribe a treatment to the patient without being present [22]. Telerehabilitation has the advantage of addressing multiple components of health, including functional independence, self-care, and disease self-management, providing patients with greater flexibility and accessibility to health services. Some disadvantages have also been identified, such as the fact that some patients may find it difficult to use new technologies or that the therapeutic relationship may change. Its use has shown positive clinical outcomes, leading to fewer emergency and physician visits, fewer hospital admissions, shorter hospital stays, and lower costs. In addition, it enables rehabilitation in remote geographic areas with limited services and a lack of access to physical therapy services [23,24,25].

Despite existing solid evidence that therapeutic education can help manage pain and prevent migraine episodes, most of the patients with CM receive only pharmacological treatment [26,27,28]. For this reason, the main objective of this study was to analyze whether adding a complementary telehealth education program (TTEP) to pharmacological treatment leads to a greater reduction in headache frequency in patients with CM when compared with a general health recommendation program.

## 2. Materials and Methods

### 2.1. Study Design

A randomized, double-blind, controlled pilot study with two parallel groups was conducted. All patients received the standard treatment, which consisted of an onabotulinumtoxinA (BTX-A) injection. The patients only received BTX-A during the study period. When necessary, they used NSAIDs, triptans, or a combination of NSAIDs as rescue treatments. Both the intervention and control groups received identical study actions, with the only difference being the specific intervention applied to the treatment group. This ensured that any differences in outcomes were attributable to the intervention itself. Patients in the experimental group (EG) received an asynchronous TTEP, whereas those in the control group (CG) received a program with general educational health recommendations. The effects were compared within and between the groups.

The study followed the CONSORT guidelines for randomized pilot and feasibility studies. All participants signed an informed consent form before their participation. The Aragon Ethics Committee approved the study (PI21/014).

This trial was registered on ClinicalTrials.gov (NCT04788667) and followed the clinical practice principles of the Declaration of Helsinki. This study complied with the recommendations for developing clinical trials regarding inclusion criteria, assessment measures, and the statistical analysis of the guidelines created by the International Headache Society (IHS) [28].

### 2.2. Study Participants

Patients were recruited at Lozano Blesa Clinic University Hospital in Zaragoza, Spain. The inclusion criteria were as follows: (1) patients diagnosed with CM following the ICHD-III criteria; (2) aged between 18–65 years; (3) migraine onset before the age of 50 years; (4) had been diagnosed with CM for at least 1 year; (5) were being treated with BTX-A. Exclusion criteria were as follows: (1) women who were pregnant or breastfeeding, or with menstrual migraine and (2) patients with severe or unstable psychiatric pathology.

The neurology service informed patients who attended the headache service and met the inclusion criteria about the project, and they signed the informed consent form.

The participants were assigned to the EG or CG in a 1:1 ratio via the www.randomizer.org software. An independent researcher performed the randomization. The patients, neurologists, and assessors were blinded to group allocation in this study.

### 2.3. Interventions

The intervention lasted a total of 8 weeks and was delivered via a telehealth application (HEFORA app or web). Once the patients were recruited, they were called by the physical therapists carrying out the intervention to help them register and install the application, and to solve any questions related to the use of the HEFORA app/web platform, which was used to provide the treatment and perform the patient assessment (through questionnaires). The intervention was delivered via the HEFORA app, which provided educational content and allowed for patient assessment through questionnaires. The experimental group (EG) and the control group (CG) received the same number of video tutorials covering eight key health topics: pain education, sleep and eating habits, physical exercise, postural hygiene, relaxation techniques, and pain management. However, while the EG received 15 videos tailored explicitly to migraine management—covering topics such as migraine neurophysiology and self-management strategies—the CG received general health recommendations, without specific content related to migraine. The educational content for the EG was self-designed by the authors based on the literature on health education for chronic pain [29,30,31,32,33,34,35,36,37,38]. The content of these videos is summarized in more detail in Table 1.

During the intervention, patients were able to communicate directly with the researcher via the app/web to clarify any doubts or problems that arose. The patients watched one topic per week and the researcher checked each week that patients had completed the tests and videos. In cases of a lack of compliance, the researcher contacted them to send them the questionnaires and reminded them to ensure they watched the videos and completed the tests correctly.

### 2.4. Outcome Measurements:

Baseline data, including age, gender, body mass index, and tobacco and coffee quantity/day were collected. The primary outcome measure was the headache frequency (days/month). Headache frequency included headache days with and without migraine features in the last month. It was recorded using an electronic pain diary, created with the HEFORA app. This pain diary also included some secondary outcomes, which were migraine frequency (days/month)—a migraine day is defined as a day on which the patient has symptoms for more than 4 h according to the ICDH III classification, or on which the headache disappears within 2 h after taking specific migraine medication—and headache intensity, which was measured on a 4-point scale (0 = no pain; 1 = mild headache; 2 = moderate headache; and 3 = severe headache). The average pain per month was measured [5,28].

Rescue medication was carefully tracked and recorded throughout the study via the pain diary. Headache frequency, migraine frequency, and headache intensity were recorded for one month before the intervention (M0), one month after the start of the intervention (M1), at the end of the intervention (M2), and one month after the end of the intervention (1-month follow-up) (M3).

The other secondary outcome measures were as follows:

The headache impact (HIT-6) consisted of 6 items and it was used to measure the impact of headaches on patients’ functional abilities, work, school, and social situations. Scores ranged from 36 to 78 points, with scores above 60 indicating very severe impact and below 49 indicating low or no impact [38,39,40].

Cutaneous allodynia was measured with the Allodynia Symptom Checklist (ASC-12). This scale quantifies cutaneous allodynia globally and is based on assessing various clinical symptoms. It includes 12 questions on the frequency of various cutaneous allodynia symptoms associated with a migraine attack. The score can range from 0 to 24 and is interpreted as follows: score 0–2 (no allodynia), score 3–5 (mild allodynia), score 6–8 (moderate allodynia), and Score > or = 9 (severe allodynia) [41].

Fear of movement, as measured by the Tampa Scale of Kinesiophobia (TSK-11), is a self-administered 11-item fear of movement or injury scale. Patients should indicate the level of agreement on a Likert scale ranging from 1 = strongly disagree to 4 = strongly agree. High scores indicate a greater fear of movement [42].

Pain catastrophizing was assessed using the Pain Catastrophizing Scale (PCS), a 13-item scale in which the patient responds to each item on a 5-point Likert scale, ranging from 0 = never to 4 = always. Scores range from 13 to 62 points, with low scores indicating low catastrophizing and high scores indicating high catastrophizing [43].

Chronic pain self-efficacy was measured with the Chronic Pain Self-Efficacy Scale (PSEQ), which has 19 items, for which the respondent indicates the extent to which they consider themselves capable of carrying out specific actions. The scale is answered on a Likert scale ranging from 0 (= I see myself as totally incapable) to 10 (= I see myself as totally capable). A high score indicates a high perception of self-efficacy [44].

Anxiety and depression levels were measured with the Hospital Anxiety and Depression Scale (HADS). This questionnaire consists of two subscales, each with 7 items that can be rated on a Likert scale from 0 to 3. Anxiety are the odd items and depression the even items, and the scores in each subscale range from 0–21. The higher the score, the higher the anxiety and depression. The authors suggested that scores above eleven would indicate “case” and above eight would be considered “probable case” in both subscales [45].

Sleep quality was measured with the Pittsburgh Sleep Quality Index (PSQI), which uses 19 items to measure sleep quality, latency, duration, efficiency, sleep disturbances, and daytime dysfunctions. The score ranges from 0 to 21, with 0 equaling no difficulty and 21 equaling severe sleep difficulty [46].

A sedentary lifestyle was measured using the short version of the IPAQ questionnaire, which contains 7 questions on the frequency, duration, and intensity of physical activity performed in the last seven days and time spent sitting at work. Three aspects are evaluated: intensity, frequency, and duration. Based on the results obtained after calculating the metabolic index units (METS), patients are classified according to their level of physical activity: low, moderate, or high [47].

These secondary outcomes (headache impact, allodynia, fear of movement, catastrophizing, chronic pain self-efficacy, anxiety and depression, sleep quality, and sedentary lifestyle) were determined through self-administered questionnaires in the HEFORA app and were assessed three times: at baseline, before randomization; post-treatment (immediately after 8 weeks of intervention); and at 1-month follow-up.

### 2.5. Sample Size and Statistical Analysis

Following the recommendations for randomized controlled trial pilot studies, a sample size of 50 participants, 25 participants per arm, was determined, considering 20% for possible drop-outs [48,49].

The data were analyzed using the statistical software IBM SPSS Statistics Version 29.0 (SPSS Inc., Chicago, IL, USA). The normal distribution of the quantitative variables was tested using the Shapiro–Wilk test. Descriptive statistics were expressed as mean ± standard deviation or median [interquartile range] for continuous parameters, and as frequency (%) for categorical data. Baseline measurements were compared between groups using the independent Student’s *t*-test, the Mann–Whitney U test, and the chi-square test.

Between- and within-intervention analyses were performed using one-way ANOVA and mixed ANOVA, with Bonferroni post-hoc pairwise comparisons when a normal distribution was found. Assuming a non-normal distribution, non-parametric analyses were performed using the Mann–Whitney U test to compare interventions and the Friedman test with the Tukey post-hoc test to highlight within-intervention differences. In addition, the effect size was calculated using Cohen’s d coefficient and interpreted as small (d = 0.2), medium (d = 0.5), or large (d > 0.8) [50]. The significance level was established at *p* < 0.05. An intention-to-treat (ITT) procedure was carried out.

## 3. Results

Recruitment started in May 2022 and was completed by January 2024. In total, 50 patients were recruited. Two participants were excluded from the study because they did not correctly complete the baseline data collection for personal reasons, and 48 patients who met the selection criteria and completed the baseline data were randomly assigned to the EG or CG (Figure 1).

Both groups were similar at baseline for the different outcomes (Table 2).

There were differences between the groups in the primary endpoint, the headache frequency, and in the one-month follow-up in favor of EG (*p* = 0.03; d = 0.681), with a difference of 2.86 days (95% CI 0.28 to 5.43). Regarding the within-group analysis, the number of headache days decreased in both groups, compared to the pre-intervention measurement, and the differences were statistically significant (*p* < 0.05) in all comparisons, except for the comparison between the baseline and month one for the CG (Table 3 and Table 4).

There were also between-group differences in chronic pain self-efficacy at post-treatment (*p* = 0.007; d = 0.885), with a difference of 23.55 points (95% CI −39.80 to −7.29), and at the one-month follow-up (*p* < 0.001; d = 0.998), with a difference of 26.21 points (95% CI −41.69 to −10.74) (Table 4). There were also between-group differences in sleep quality at post-treatment (3.45; 95% CI 0.77 to 6.13) (*p* = 0.013; d = 0.786) and at the one-month follow-up (4.27; 95% CI 1.86 to 6.67) (*p* < 0.001; d = 1.086) (Table 5). There were no differences between the groups in the other outcomes assessed (*p* < 0.05).

In the within-group analyses, a statistically significant improvement in the frequency of migraine was observed in the EG after 1 month (*p* < 0.001; d = 0.980), at post-treatment (*p* < 0.001; d = 0.878), and at follow-up (*p* < 0.001; d = 1.080). The CG did not change significantly over time (*p* > 0.05) (Table 3). The EG also had statistically significant improvements in allodynia at post-treatment (*p* = 0.002, d = 0.325) and at the one-month follow-up (*p* < 0.001; d = 0.483); in kinesiophobia at post-treatment (*p* < 0.001; d = 0.580) and at the one-month follow-up (*p* < 0.001; d = 0.713); and in anxiety at post-treatment (*p* < 0.001; d = 0.915) and at the one-month follow-up (*p* < 0.001; d = 1.336). The CG only showed significant improvements at the one-month follow-up for allodynia (*p* = 0.002; d = 0.323), kinesiophobia (*p* < 0.001; d = 0.399), and anxiety (*p* = 0.045; d = 0.235) (Table 5 and Table 6).

Furthermore, both the EG and CG had statistically significant improvements over time in pain catastrophizing at both post-treatment (*p* < 0.001, d = 0.880 for the EG; *p* < 0.001, d = 0.918 for the CG) and at the one-month follow-up (*p* < 0.001, d = 0.637 for the EG; *p* < 0.001, d = 0.711 for the CG); in depression at both post-treatment (*p* = 0.018, d = 0.256 for the EG; *p* = 0.034, d = 0.228 for the CG) and at the one-month follow-up (*p* < 0.001, d = 0.415 for the EG; *p* < 0.001, d = 0.427 for the CG); and in the level of physical activity at both post-treatment (*p* < 0.001, d = 0.462 for the EG; *p* = 0.06, d = 0.184 for the CG) and at the one-month follow-up (*p* < 0.001, d = 0.459 for the EG; *p* = 0.003, d = 0.186 for the CG) (Table 5 and Table 6). No significant changes were found in pain intensity or headache impact for either group (*p* > 0.05) (Table 4 and Table 5).

There were no differences between the groups with regard to the use of rescue medication (*p* > 0.05).

## 4. Discussion

This was the first study to analyze whether adding a TTEP to pharmacological treatment could be a more effective approach to the reduction of headache frequency in patients with CM, compared to a general health recommendation program. The analysis between the groups seemed to indicate that the TTEP, which was used for the EG, was indeed preliminarily more effective in reducing headache frequency and improving several secondary outcomes, compared to the CG, who were receiving general health recommendations.

It is important to note that there is no evidence in the literature of studies comparing different types of education for migraine patients and integrating them into pharmacological treatment. Therefore, directly comparing our results with previous studies is not feasible. However, some studies have compared pain neuroscience interventions that have used general education or health recommendations for other disorders. For instance, a systematic review by Lepri et al. found that specific pain neuroscience education is more effective than general education in improving pain, disability, and psychosocial factors in patients with chronic musculoskeletal pain and central sensitization [51]. Additionally, Suso-Martí et al. reported that general education might not be specific enough to significantly influence a patient’s pain experience. These findings were consistent with our study’s results, which indicated improvements in headache frequency, pain self-efficacy, and sleep quality in the EG receiving the tailored TTEP [52].

Regarding headache frequency, significant differences between the groups were found in favor of the EG at the one-month follow-up. For migraine frequency, a significant trend was also found in the EG one-month after the start of the intervention, with no significant difference after treatment or in the follow-up. The systematic review by Minen et al. showed that education in pain neuroscience and coping with pain has emerged as a promising therapy, reducing headache and migraine days when used along with traditional pharmacological treatments [29]. Our results were also consistent with Thakur’s study, which reported a significant reduction in the frequency and intensity of migraine episodes in patients who were provided with an online migraine educational video [53]. CM has been associated with persistent cortical changes, including the anterior cingulate cortex (ACC), anterior insula, and postcentral gyrus, which play a crucial role in the affective perception of pain and are also associated with headache frequency [54,55,56,57]. The study by Fedeli et al., in which patients with migraine were assigned to the mindfulness plus usual care group or to the usual care group, also reported a significant reduction in headache frequency (*p* = 0.028) and an increase in the functional connectivity of the nervous system in the mindfulness group. This was due to changes primarily in the ACC through the modulation of pain-processing areas [58]. These results can be related to ours, in which the ACC could have been modulated by the cognitive and emotional skills learned through the TTEP. These skills enable patients to better manage their pain, which could influence CCA activity, thereby reducing headache frequency and perceived pain intensity and improving their overall well-being [55].

Self-efficacy in managing chronic pain is crucial as it helps to reduce pain and improve functionality. High self-efficacy is associated with better coping strategies, including better emotional regulation. It has also been shown that people with effective coping strategies are better able to control stress and anxiety. In this study, significant differences were found between the groups after receiving education in coping strategies for chronic pain, both after treatment and after a the one-month follow-up [59,60,61]. As mentioned above, the CCA is involved in attention to pain and plays a key role in emotional pain regulation [62]. Increased self-efficacy may reduce the perceived threat of pain, which may decrease the CCA’s activation in response to pain. Therefore, the improvements in self-efficacy achieved in this study could have facilitated better regulation of the negative emotions associated with pain, thereby modulating CCA activity. These findings were consistent with the study by Bromberg et al., in which a web-based intervention on migraine, coping strategies, and self-management techniques was conducted, and significant improvements in self-efficacy and migraine frequency were achieved [63].

Regarding sleep quality, a significant improvement was observed in favor of the EG, both post-treatment and at the one-month follow-up. These results could be relevant since there is an association between poor sleep quality and migraine, as well as the frequency of migraine attacks. In fact, 85.41% of patients in this study had sleep disturbances at baseline. Other studies have shown the importance of modifying the factors of sleep habits that facilitate rest to improve the quality of life of migraine patients [64,65]. The results obtained in the headache and migraine frequency outcomes may have contributed to the improvement in sleep quality. Also, the results related to anxiety, in which a significant trend was found in the EG after treatment and at the one-month follow-up, may have improved sleep quality by reducing stress and managing depression through the TTEP received, which included techniques such as learning to cope with stressors, problem solving, lifestyle changes, and exercise, in accordance with evidence-based recommendations [66,67].

Although there were no differences between the groups, the EG had significant improvements in cutaneous allodynia and kinesiophobia after treatment and at the one-month follow-up. The CG also showed significant improvements in the same variables, but only at the one-month follow-up. These results suggest that, while general health recommendations may have some impact, the TTEP provides a more targeted approach, which better addresses the psychological and behavioral aspects of chronic migraine. Pain catastrophizing and cutaneous allodynia represent two risk factors for greater headache-related disability [68,69]. Farris et al., who assigned patients to either 16 weeks of behavioral weight loss therapy or migraine education, also found a significant reduction in pain catastrophizing and allodynia for both groups in both the post-treatment and follow-up [70]. The changes in the CG may have been due to the fact that the CG also received general health recommendations that, while not migraine-specific, may have had an impact on changing the behavioral habits that led to the observed long-term improvements. These results are consistent with the literature, where cognitive behavioral interventions for migraine produce improvements in pain catastrophizing, although they do not directly target pain catastrophizing. As in the study by Farris et al., our study also showed that the neuropsychological aspects of pain sensitivity decrease after migraine education intervention, which is a novel finding that needs to be investigated in future studies [63,70,71].

Both groups reported improvements in depression and physical activity after treatment and at the one-month follow-up, although the EG showed a sizeable clinical effect after treatment and at the one-month follow-up. These results are aligned with Kindelan-Calvo’s review, which reported that, although therapeutic patient education appears to be an effective tool for reducing the symptoms of depression, according to their findings, it does not have much impact [21,72]. In contrast, Bromberg et al. found an improvement in depression in the EG receiving a web-based intervention [63]. Coping with negative cognitions or negative emotions (depression) are key factors that help people gain a sense of control, and they are an important predictor of quality of life in people with migraine headaches. The fact that both groups improved could have been because our study design was a double-blind, placebo-controlled trial; it has been noted by various researchers in the field of migraine that it is often complicated when testing a behavioral intervention to create a “behavioral placebo”, as the Hawthorne effect may occur (improvement due to being studied and receiving the researchers’ attention). Some participants may have improved over time regardless of the intervention administered [73].

Despite finding differences between the groups in the primary outcome, this pilot study had some limitations. One such limitation was the potential lack of power in detecting differences in some secondary outcomes, as changes were observed over time in the experimental group. However, significant differences between the groups were found in chronic pain self-efficacy and sleep quality. Another possible reason for the absence of group differences in other secondary outcomes could be the intervention’s limited duration of eight weeks, which may not have been sufficient to produce a significant effect. Additionally, adherence to the treatment was not measured. These limitations could be addressed in a larger clinical trial, with an increased sample size, which may clarify whether the observed effects in some secondary measures were real but undetected or whether the intervention was effective only for certain outcomes.

## 5. Conclusions

This pilot study suggests that adding a complementary therapeutic telehealth education program to the pharmacological treatment may be more effective in the preventive treatment for migraine than providing general health recommendations. The addition of targeted education to the pharmacological treatment of CM appears to reduce headache frequency, improve pain self-efficacy and sleep quality, and show a positive trend in improving the number of migraine days experienced by patients. These findings highlight the need for further research, with larger sample sizes to validate the effectiveness of this approach. This study has paved the way for future research to better understand the critical role of selecting the appropriate educational interventions for migraine patients.

## Figures and Tables

**Figure 1 jcm-13-06825-f001:**
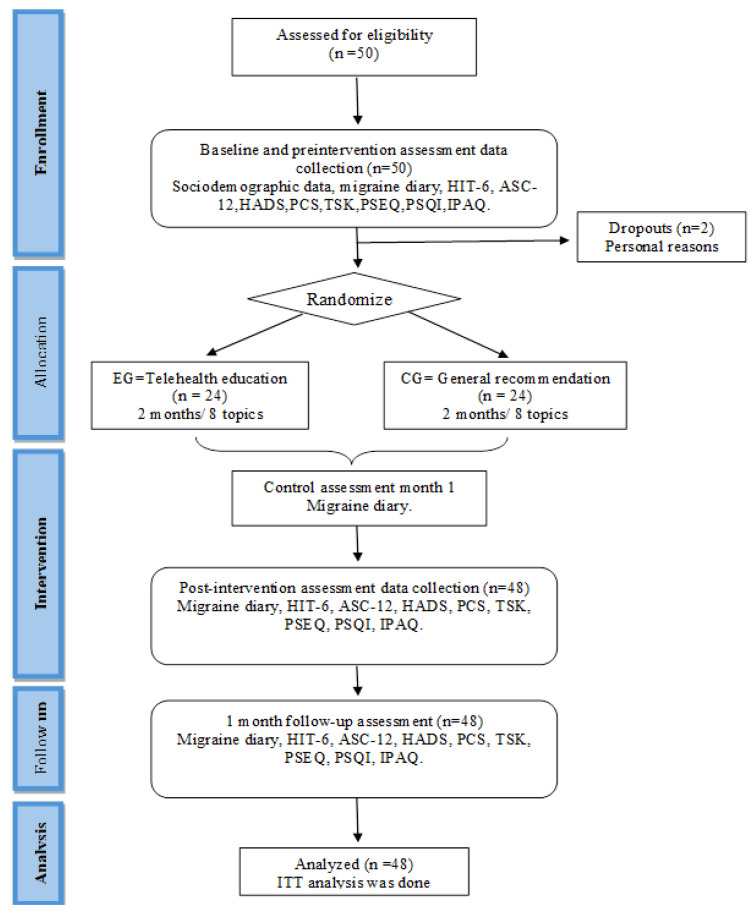
CONSORT flow diagram.

**Table 1 jcm-13-06825-t001:** Topics for the videos in health and pain education.

Video	Topic	Content
1	Pain neuroscience	What is pain? How does it happen?
2	Pain neuroscience	What is chronic pain and what are the implications of chronic pain?
3	Migraine	What is migraine?
4	Migraine	What is chronic migraine? Why does it happen?
5	Migraine	Risk factors that may affect migraine attacks.
6	Pain management	Non-drug strategies to reduce the number and intensity of seizures.
7	Sleeping habits	Strategies to improve sleep quality and rest.
8	Eating habits	General recommendations on eating habits
9	Eating habits	Foods that can make migraine attacks better/worse.
10	Eating habits	False myths about miracle diets and which diets are effective in controlling migraine attacks.
11	Physical exercise	General exercise recommendations.
12	Physical exercise	How to perform aerobic exercise.
13	Physical exercise	How to perform strength–endurance exercise.
14	Postural hygiene	Understanding posture and how it can affect migraines.
15	Relaxation techniques	Effects of relaxation on pain and different relaxation strategies.

**Table 2 jcm-13-06825-t002:** Baseline demographic and clinical characteristics.

	EG	CG	Between Groups-Differences
Age (years) (m ± SD)	43.21 ± 7.35	43.15 ± 11.19	0.984 ^¥^
Sex (% female)	95.83%	100%	0.356 ^ƶ^
BMI (kg/cm^2^)			
	4.16%%	0%	0.571 ^ƶ^
Underweight (%)	58.33%	45.00%	
Normal Weight (%)	29.16%	40.00%	
Overweight (%)	8.33%	15.00%	
Obesity (%)			
Headache frequency (days/month)	18.87 ± 5.14	18.55 ± 3.94	0.818 ^¥^
Migraine frequency (days/month)	15.79 ± 7.24	14.45 ± 4.94	0.486 ^¥^
Migraine intensity	0.502 ^ƶ^		
Mild (1)		35.09%	21.02%
Moderate (2)		36.42%	42.58%
Severe (3)		28.25%	36.11%
Smoking (% yes)	25.00%	25.00%	0.711 ^ƶ^
Coffee (% yes)	70.83%	50.00%	0.450 ^ƶ^
Headache Impact Test (HIT-6)	64 (0–78)	0 (0–76)	0.26 ^‡^
Cutaneous Allodynia (ASC-12)	4.5 (0–19)	11 (0–20)	0.27 ^‡^
Tampa Scale of Kinesiophobia (TSK-11)	26.5 (18–44)	29 (17–44)	0.59 ^‡^
Pain Catastrophizing Scale (PCS)	31.00 ± 11.07	29.75 ± 12.56	0.72 *
Chronic pain self-efficacy scale	97.70 ± 30.75	93.80 ± 34.72	0.69 *
Anxiety (HADS)	9.54 ± 4.46	9.45 ± 5.28	0.72 *
Depression (HADS)	9 (1–21)	6 (1–21)	0.51 ^‡^
Sleep disorders (PSQI)	12.54 ± 4.99	12.50 ± 4.38	0.57 *
IPAQ (METS/week)	883.5 (99–3306)	765.25 (198–9234)	0.85 ^‡^

^¥^ Using independent *t*-test; ^ƶ^ using chi-squared test; * using one-way ANOVA; ^‡^ using Mann–Whitney U test.

**Table 3 jcm-13-06825-t003:** Within group comparisons of the pain diary outcomes at baseline, one month after starting the treatment, post-treatment, and follow-up.

Variable		Descriptive Data				Within-Group Effect					
		Baseline (M0)	Month 1 (M1)	Post-Treatment (M2)	Follow-Up (M3)	M1 vs. M0		M2 vs. M0		M3 vs. M0	
		Mean ± SD	Mean ± SD	Mean ± SD	Mean ± SD						
		Median (IQR)	Median (IQR)	Median (IQR)	Median (IQR)	*p* Value	Effect Size	*p* Value	Effect Size	*p* Value	Effect Size
Frequency of headache (days of headache)	EG	18.87 ± 5.14	14.08 ± 4.11	13.75 ± 4.41	12.54 ± 4.25	<0.001 ^†^	**0.68**	**<0.001 ^†^**	**1.161**	**<0.001 ^†^**	**1.489**
		18 (10–29)	14 (8–23)	13 (6–24)	(13 (7–23)						
	CG	18.55 ± 3.94	16.35 ± 3.73	15.00 ± 3.09	15.40 ± 4.15	0.059 ^†^	0.59	**0.002 ^†^**	**1.149**	**0.013 ^†^**	**0.759**
		18 (12–25)	17 (11–27)	15 (9–21)	15.5 (9–28)						
Frequency of migraine (days of migraine)	EG	15.79 ± 7.24	10.83 ± 5.06	10.33 ± 6.22	10.66 ± 4.75	**<0.001 ^†^**	**0.98**	**<0.001 ^†^**	**0.878**	**<0.001 ^†^**	**1.08**
		15 (0–29)	10.5 (0–21)	10 (0–24)	10 (5–23)						
	CG	14.45 ± 4.94	13.85 ± 4.86	13.35 ± 4.46	12.70 ± 5.93	0.999 ^†^	0.123	0.999 ^†^	0.247	0.999 ^†^	0.295
		13.5 (8–24)	13 (5–26)	13.5 (1–20)	13 (1v28)						
Intensity of pain	EG	1.63 ± 0.37	1.70 ± 0.33	1.64 ± 0.34	1.73 ± 0.33	0.990 ^†^	0.212	0.990 ^†^	0.029	0.999 ^†^	0.303
		1.55 (1.2–2.7)	1.60 (1.3–2.5)	1.65 (1.2–2.5)	1.73 (1.2–2.5)						
	CG	1.92 ± 0.33	1.80 ± 0.30	1.75 ± 0.40	1.85 ± 0.39	0.310 ^†^	0.4	0.290 ^†^	0.425	0.990 ^†^	0.179
		1.89 (1.2–2.4	1.87 (1.1–2.3)	1.66 (1.0–2.5)	1.93 (1.0–2.4)						

^†^ Using mixed-design ANOVA. Significant *p*-values and effect sizes are in bold.

**Table 4 jcm-13-06825-t004:** Between-group comparison of the pain diary outcomes one month after starting the treatment, post-treatment, and follow-up.

Variable	Between-Group Effects					
	Month 1 (M1)		Post-Treatment (M2)		Follow-Up (M3)	
	*p* Value	Effect Size	*p* Value	Effect Size	*p* Value	Effect Size
Frequency of migraine (days of migraine)	0.065 *	0.578	0.290 *	0.328	0.030 *	0.681
Frequency of migraine (days of migraine)	0.052 *	0.609	0.077 *	0.558	0.210 *	0.38
Intensity of pain	0.320 *	0.317	0.300 *	0.299	0.270 *	0.335

* Using one-way ANOVA. Significant *p*-values and effect sizes are in bold.

**Table 5 jcm-13-06825-t005:** Within- and between-group comparison of the secondary outcomes at baseline, post-treatment, and follow-up.

Descriptive Data					Within-Group Effects				Between-Group Effects			
Variable		Baseline (Mean ± SD) Median (IQR)	Post-Treatment (Mean ± SD) Median (IQR)	Follow-Up (Mean ± SD) Median (IQR)	Post-Treatment vs. Pre-Treatment		Follow-Up vs. Pre-Treatment		Post-Treatment		Follow-Up	
					*p* Value	Effect Size	*p* Value	Effect Size	*p* Value	Effect Size	*p* Value	Effect Size
Headache Impact Test (HIT-6)	EG	39.33 ± 37.12	37.08 ± 35.22	36.58 ± 35.08	0.580 ^§^	0.064	0.870 ^§^	0.078	0.780 ^‡^	0.227	0.490 ^‡^	0.257
		64 (0–78)	56 (0–76)	50 (0–78)								
	CG	27.75 ± 35.33	28.95 ± 36.57	27.55 ± 35.10	0.220 ^§^	0.033	0.240 ^§^	0.006				
		0 (0–76)	0 (0–78)	0 (0–78)								
Cutaneous Allodynia (ASC-12)	EG	6.12 ± 4.72	4.87 ± 3.85	4.37 ± 3.62	**0.002 ^§^**	**0.325**	**<0.001 ^§^**	**0.483**	0.190 ^‡^	0.58	0.200 ^‡^	0.558
		4.5 (0–19)	3.5 (1–15)	3 (1–14)								
	CG	8.45 ± 6.01	7.65 ± 5.73	6.80 ± 5.11	0.180 ^§^	0.14	**0.002 ^§^**	**0.323**				
		11 (0–20)	8.5 (0–20)	8 (0–17)								
Tampa Scale Kinesiophobia (TSK-11)	EG	28.91 ± 7.59	24.87 ± 6.98	23.50 ± 6.67	**<0.001 ^§^**	**0.58**	**<0.001 ^§^**	**0.713**	0.972 ^‡^	0.046	0.653 ^‡^	0.142
		26.5 (18–44)	23.5 (15–40)	23 (14–39)								
	CG	27.80 ± 8.56	25.20 ± 7.27	24.85 ± 7.40	0.066 ^§^	0.358	**<0.001 ^§^**	**0.399**				
		29 (17–44)	26.5 (15–37)	25.5 (14–38)								
Pain Catastrophizing Scale (PCS)	EG	31.00 ± 11.07	23.45 ± 8.58	22.70 ± 9.04	**<0.001 ^†^**	**0.88**	**<0.001 ^†^**	**0.918**	0.960 *	0.011	0.910 *	0.031
		29.5 (13–49)	22 (12–42)	21 (10–44)								
	CG	29.75 ± 12.56	23.35 ± 10.05	22.40 ± 10.34	**<0.001 ^†^**	**0.637**	**<0.001 ^†^**	**0.711**				
		31 (10–52)	22 (9–42)	20.5 (8–44)								
Chronic pain self-efficacy scale (PSEQ)	EG	97.70 ± 30.75	123.75 ± 22.94	128.16 ± 22.22	**<0.001 ^†^**	**1.136**	**<0.001 ^†^**	**1.371**	**0.007 ***	**0.885**	**<0.001 ***	**0.998**
		103 (21–154)	128 (68–170)	130 (72–172)								
	CG	93.80 ± 34.72	100.20 ± 30.45	101.95 ± 28.63	0.291 ^†^	0.24	**0.006 ^†^**	**0.285**				
		99 (20–139)	107.5 (34–142)	110 (45–136)								

^†^ Using mixed-design ANOVA; ^§^ using Friedman test; * using one-way ANOVA; ^‡^ using Mann–Whitney U test. Significant *p*-values and effect sizes are in bold.

**Table 6 jcm-13-06825-t006:** Within- and between-group comparison of the secondary outcomes at baseline, post-treatment, and follow-up.

Descriptive Data					Within-Group Effects				Between-Group Effects			
Variable		Baseline (Mean ± SD) Median (IQR)	Post-Treatment (Mean ± SD) Median (IQR)	Follow-Up (Mean ± SD) Median (IQR)	Post-Treatment vs. Pre-Treatment		Follow-Up vs. Pre-Treatment		Post-Treatment		Follow-Up	
					*p* Value	Effect Size	*p* Value	Effect Size	*p* Value	Effect Size	*p* Value	Effect Size
Anxiety (HADS)	EG	9.54 ± 4.46	6.54 ± 3.28	6.00 ± 2.65	**<0.001 ^†^**	**0.915**	**<0.001 ^†^**	**1.336**	0.083 *	0.538	0.083 *	0.538
		9 (2–19)	6.5 (1–13)	6 (1–11)								
	CG	9.45 ± 5.28	8.95 ± 5.60	8.20 ± 5.33	0.725 ^†^	0.089	**0.045 ^†^**	**0.235**				
		8.5 (2–19)	9.5 (1–20)	8.5 (1–19)								
Depression (HADS)	EG	10.16 ± 6.78	8.66 ± 5.85	7.91 ± 5.42	**0.018 ^§^**	**0.256**	**<0.001 ^§^**	**0.415**	0.390 ^‡^	0.155	0.280 ^‡^	0.235
		9 (1–21)	7 (2–20)	7 (1–20)								
	CG	9.20 ± 7.11	7.70 ± 6.59	6.55 ± 6.21	**0.034 ^§^**	**0.228**	**<0.001 ^§^**	**0.427**				
		6 (1–21)	5.5 (1–20)	4 (1–20)								
Sleep disorders (PSQI)	EG	12.54 ± 4.99	8.50 ± 4.14	7.33 ± 3.67	**<0.001 ^†^**	**0.976**	**<0.001 ^†^**	**1.42**	**0.013 ***	**0.786**	**<0.001 ***	**1.086**
		14 (4–19)	7.5 (2–16)	6.5 (2–15)								
	CG	12.50 ± 4.38	11.95 ± 4.67	11.60 ± 4.23	0.573 ^†^	0.118	0.213 ^†^	0.213				
		13.5 (3–19)	13.5 (3–18)	12 (3–18)								
IPAQ (METS/week)	EG	1079.39 ± 884.44	1420.45 ± 738.77	1420.45 ± 743.51	**<0.001 ^§^**	**0.462**	**<0.001 ^§^**	**0.459**	0.430 ^‡^	0.147	0.410 ^‡^	0.149
		883.5 (99–3306)	1306.25 (99–3306)	1306 (99–3306)								
	CG	1465.37 ± 1994.07	1627.75 ± 1927.10	1631.75 ± 1944.63	**0.006 ^§^**	**0.184**	**0.003 ^§^**	**0.186**				
		765.25 (198–9234)	1005 (297–9234)	1005 (297–9234)								

^†^ Using mixed-design ANOVA; ^§^ using Friedman test; * using one-way ANOVA; ^‡^ using Mann–Whitney U test. Significant *p*-values and effect sizes are in bold.

## Data Availability

The datasets generated during and/or analyzed during the current study are available from the corresponding author upon reasonable request.

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
