# Peer review of "Effectiveness of a Complementary Telehealth Education Program as a Preventive Treatment for Chronic Migraine: A Randomized Pilot Study"

_jcm, 2024, doi:10.3390/jcm13226825_

Round 1
Reviewer 1 Report
Comments and Suggestions for Authors
Dear Authors,
Thank you for taking on this important topic. I thought your study was well designed and researched. The conclusions are supported by the results. It is extremely important that such a relatively easy intervention should result in clear improvement in CM. As you correctly point out this is a small sample size, but the findings are important and relevant.
Author Response
Comment 1: Thank you for taking on this important topic. I thought your study was well designed and researched. The conclusions are supported by the results. It is extremely important that such a relatively easy intervention should result in clear improvement in CM. As you correctly point out this is a small sample size, but the findings are important and relevant.
Reply 1: We would like to thank you for reviewing our manuscript.
Reviewer 2 Report
Comments and Suggestions for Authors
Dear authors,
The topic is interesting in clinical practice for management of chronic migraine (CM) patients. Overall, the writing and presentation gained moderate clarity, intectual concept and rationales of the intervention done.
My suggestions and concerns were as follows:
1. Main concept and rationale of the study.
It has been known that adding an eductional program (EP) is useful in management a chronic disable pain condition. What I would like the authors to clarify more in details include:-
1.1 Why were only CM patients who were treating with botulinumtoxin-A enrolled in this study?
Whether other groups of migraine patients will not gain benefit from this intervention.
1.2 The topics of EP selected in this study were based on specific scientific recommendations for this specific group of CM patients, if so please address in Introduction, or self-designed by the authors.
1.3 What did the neuro-physiological, health psychological or behavioro-social concepts support the 15 topics selected to form the EP? Please review the relevant literatue to strengthening their potential useful application in CM patients of this study setting in Introduction. Also, the previous studies reporting positive outcomes of the topics in CM patients should be emphasized.
1.4 Why did tele-health method selected for EP provision? Were there any advantages over usual face-to-face knowledge provision method? Were any potential disadvantages possibly encountered?
1.5 And, how did the advantage vs. disadvantage factors affect the study outcome. Please the authors discuss this issue.
1.6 Did the participants consistenly maintain the original anti or prophylatic pharmacological treatment regimen for migraine through the period of 8-week study time. Were any additional abortive drugs or supportive treatments added for rescuing acute pain?
2. Introduction: Expand the content by more extensive review of the published papers regarding this topic of concern. I just only a few lines of the expected content at the last pargraph of Introduction.
3. Study Methods
The description of study methods needs more clarification in steps for the reader to follow and understand, e.g
3.1 The authors showed in the Consort Diagram that 8 topics in the arm of study, and 8 topics in control. Were they the topics early mentioned in the text, and how, or what were the reasons, were they distributed to each arm? If it would be better to address the topics in individual arm in the Consort Diagram.
3.2 How did the authors assure that the participants completely comply to the study protocol. What was the monitoring system for the compliance?
3.3 As the evaluation method done 1 month after intervention at a follow-up was self-reported by the participants, Could this deviate the results? What supportive actions should be added for high precision.
3.4 The duration of the intervention was 8 week, in which the evaluations were done at baseline (M0), one-month after starting intervention (M1), at the intervention end (M2), and 1 follow-up month after completing intervention (M3) [line 135-137]. Is my understanding correct? Please adress this time points in the Abstract clearly.
3.5 The authors should emphasize that both arms (study patients and controls) received the identical study actions throughout the process to ensure the case-control study character.
4. Results: The tables in the Results may be too difficult to be read. Please modified the thick lines and improve the Table alignment. Please consider to remove some of them with less significant mening but express them as plain text instead.
5. Discussion: Overall, it is moderately fine according to the current content. After revision and more details obtained, however, it should be reconsidered its logical relevance with the foregoing parts of manuscript.
6. Minors
6.1 Fine English writing. Please recheck the spelling and puntuations errors , if any.
6.2 'APP' in the abstract stands for what ( application?, I guess)
Comments on the Quality of English Language
Fine, no serious errors to be concerned.
Author Response
Please find document attached

Reviewer 3 Report
Comments and Suggestions for Authors
General comment: The aim of this study was to analyze whether adding a therapeutic telehealth education program to pharmacological treatment achieves a greater reduction in the number of headache days in patients with chronic migraine.
Specific comments:
Summary: There is a lot of material and methods and little in terms of results.
Introduction: The introduction is succinct, but addresses the essentials, is synthetic in centralizing the problem.
Methodology: The methodology is complex: Many interventions (independent variables); Many outcomes (dependent variables); Data collection instruments explained and clarified; Justification of sample size; Data analysis methods; Clarification of ethical aspects.
Results: Table 2 is ok, as is the description of the results it expresses. The following tables 3, 4 and 5 seem very confusing to me, and the comments on them are too simplistic, the analysis is only univariate. I believe that in a study of this nature, where there are so many independent and dependent variables. The authors could use other analysis strategies, simplifying the presentation of their results, possibly using multiple comparison analysis and linear and/or simple and multiple logistic regression models. In the presence of so many variables under analysis, collinearity studies should have been carried out.
Discussion/conclusion: The authors point out possible causes for the trends found and compare them with the existing literature on the topic, analyzing and interpreting the results. The authors recognize some important limitations of their study, but there may be others, particularly related to data analysis.
Conclusion: The conclusion is a little pretentious, I don't know if the authors can draw these conclusions, based on what I mentioned in the results chapter (only univariate analysis).
Author Response
Please find document attached

Round 2
Reviewer 2 Report
Comments and Suggestions for Authors
Dear authors,
Thank you very much for your responses addressing the points of concerns and suggestions from the first review. However, to make the manscript more clear for practical implication, plese let me add more comments for further explannations and/or revisions.
General comments
In this study, the authors intended to suggest
1) The usefulness of education program added to conventional pharmacologic prophylatic treatments for CM (Botulinum toxin A).
2) Telehealth or on-line health education as an effective route of the educational content delivery to the patients.
However, for me, I consider they were not strongly addressed in the introduction and discussion to adequately convince the readers. (Please see item 3 of comments on specific parts).
Comments on the specific parts
1. Title. I suggest the word 'therapeutic' should be replaced by 'complemmentary' or 'additional' since I consider it is not a main treatment modality for CM, but only help the main treatment to result in more favorable effects when it is added.
2. Abstarct. Significant information regarding the study method, e.g. participant inclusion-exclusion criteria, randomization technique, duration of intervention applied (8 weeks), and statistical anaalyses were missed. Please explicitly brief the material and methods section of the full manuscript and contain the brief content in the abstract.
In 'conclusion' section of abstract. I suggest to alternate the positions of the two sentences in this subsection to highlight your findings first, thence, the possible effectiveness of implication this approach (TTEP).
3. Introduction. Please the authors expand the literature review by citing more previous studies about 1) the usefulness of a similar eduction program as a co-treatment for CM 2) the adventages, or possible weakness - if any, of telehealth education. The already- known basic theoretical knowledge or concepts should be just succinctly mentioned.
For the paragraph talking about "Telerehabilition..." (line 77-83) leads to somewhat confusion to the readers. How is it relevant to the context of this manuscript? To my finding from Wikipedia, it stated that "Telerehabilitation (or e-rehabilitation[1][2][3] is the delivery of rehabilitation services over telecommunication networks and the internet. Telerehabilitation allows patients to interact with providers remotely and can be used both to assess patients and to deliver therapy. Fields of medicine that utilize telerehabilitation include: physical therapy, occupational therapy, speech-language pathology, audiology, and psychology. Therapy sessions can be individual or community-based. Types of therapy available include motor training exercises, speech therapy, virtual reality, robotic therapy, goal setting, and group exercise (https://en.wikipedia.org/wiki/Telerehabilitation)
Please the authors revise or provide more details for clarification.
3. Methods.
Please describe whether the both groups received only botulinum toxin A (BTx-A) during the study. Were other medicine regularly used by the patients retained and combined to Btx-A.? Were the the regular medicines used in both group used comparable in classifications (abortive/ prophylaxis). To assure that there was no confounding effect from regular medicines used, please give the details.
The authors please add more information about:- How frequent, i.e. times/week ,did the participant have to contact or perform self-study with the telehealth intervention of this study. Again, how to confirm their regularity of compliance to the intervention applied.
4. Results. It is now acceptable overall.
How frequent were the recue abortive treatments required during intervention?, and was there any significant difference between both? Please the authors provide the data. This is considered significant in evaluating of the efficacy of tha additional educational program.
The last paragraph of Results (after Table 6), and wherever in the similar pattern, please don't read the the tables out as plain text because of causing redundancy.
5. Discussion. Generally favorable, but only rearranging the order of the content presented to improve the logical relation and fluency of readers' thinking flow.
Some minor spelling errors have been found, please the author s recheck.

Fine, only minor errors were found.
Author Response
We have attached a detailed reply to your comments in pdf

Reviewer 3 Report
Comments and Suggestions for Authors
I have no comments to make other than those I have already made.
The authors have made improvements and justified positions where they could not.
Author Response
Thanks for the review